# VIDLANKD: Improving Language Understanding via Video-Distilled Knowledge Transfer

**Zineng Tang    Jaemin Cho    Hao Tan    Mohit Bansal**
UNC Chapel Hill
{terran, jmincho, haotan, mbansal}@cs.unc.edu

## Abstract

Since visual perception can give rich information beyond text descriptions for world understanding, there has been increasing interest in leveraging visual grounding for language learning. Recently, vokenization [68] has attracted attention by using the predictions of a text-to-image retrieval model as labels for language model supervision. Despite its success, the method suffers from approximation error of using finite image labels and the lack of vocabulary diversity of a small image-text dataset. To overcome these limitations, we present VIDLANKD, a video-language knowledge distillation method for improving language understanding. We train a multi-modal teacher model on a video-text dataset, and then transfer its knowledge to a student language model with a text dataset. To avoid approximation error, we propose to use different knowledge distillation objectives. In addition, the use of a large-scale video-text dataset helps learn diverse and richer vocabularies. In our experiments, VIDLANKD achieves consistent improvements over text-only language models and vokenization models, on several downstream language understanding tasks including GLUE, SQuAD, and SWAG. We also demonstrate the improved world knowledge, physical reasoning, and temporal reasoning capabilities of our model by evaluating on the GLUE-diagnostics, PIQA, and TRACIE datasets. Lastly, we present comprehensive ablation studies as well as visualizations of the learned text-to-video grounding results of our teacher and student language models.[1]

## 1 Introduction

Language learning can be aided by grounded visual cues, as they provide powerful signals for modeling a vastness of experiences in the world that cannot be documented by text alone [5; 29; 4]. While the recent trend of large-scale language model pretraining indirectly provides some world knowledge from text, most large text corpora (e.g., Wikipedia) do not provide enough multi-modal grounding information. Previous works have explored multiple ways of grounding language to visual information such as constructing a common vector space [38; 7] and supervising the model with token-wise generated vision labels [68]. However, the widely-used image-text datasets (e.g., MS COCO [48]) are much smaller than text-only corpora in terms of word counts and vocabulary diversity for language learning.

The recent method of 'vokenization' [68] is a promising initial step towards addressing this problem by supervising language models with weakly-aligned vision-language groundings. Firstly, an image-text matching model retrieves a corresponding image to each text token in a sentence. Then a language model learns to predict the selected image (called 'voken') for each text token. This can be seen as a knowledge distillation (KD) process [33] from a vision-language grounding model to a language

---

[1]Code and models: https://github.com/zinengtang/VidLanKD

35th Conference on Neural Information Processing Systems (NeurIPS 2021), virtual.

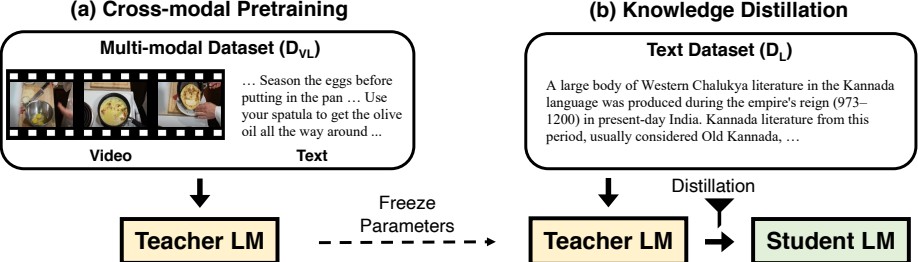

Figure 1: Overview of the proposed VIDLANKD method. We first pretrain a teacher language model on a multi-modal dataset (Sec. 3.2). Then we distill the knowledge of the teacher model (weights frozen) to a student language model on a text dataset (Sec. 3.3).

model. Although the voken classification task helps the language model to improve on natural language understanding (NLU) tasks, there exist several limitations: (1) images cannot faithfully convey word meanings that require more activity-based and physical commonsense knowledge. (2) the voken supervision suffers from approximation/quantization error of the text-to-image retrieval.

To address these problems, we propose a novel Video-and-Language Knowledge Distillation method, named VIDLANKD. Our teacher model consists of a video encoder and a language encoder. They are jointly trained with a video-language contrastive learning objective and a masked language modeling (MLM) objective on a multi-modal dataset (see Fig. 1). Then, we transfer the knowledge of the frozen teacher language encoder to a student language model by minimizing the distance between contextualized text representations of two models on a text dataset. For this, we propose to use different KD objectives including neuron selectivity transfer (NST) [34] and contrastive representation distillation (CRD) [71] that avoid the approximation error from voken assignments [68]. For cross-modal pretraining of our teacher model, we use HowTo100M [54], a large-scale video dataset which has more diverse vocabulary and richer world commonsense (e.g., physical and temporal) knowledge compared to MS COCO image dataset.

In our experiments, student language models learned with the proposed video-language KD objectives outperform the baseline text-pretrained language models and the models distilled with vokenization, on several diverse natural language understanding benchmarks including GLUE [73], SQuAD [61], and SWAG [79]. We also show comprehensive ablation studies on video encoders, student KD objectives, teacher pretraining objectives, and video vs. image-based pretraining. Furthermore, we empirically illustrate that our model successfully learns linguistic world knowledge and physical/temporal commonsense abilities from video, by showing improved performances on the GLUE-diagnostics [73], PIQA [6], and TRACIE [82] datasets.

Overall, our contributions are: (1) a novel cross-modal knowledge distillation method for improving natural language understanding, (2) using rich video-text data which can overcome the limitations of image vokenization, (3) empirical improvements on several language understanding benchmarks and studying different knowledge distillation methods, and (4) analysis on linguistic/physical/temporal knowledge learned from videos and ablation studies on the effectiveness of proposed components.

## 2 Related Work

### 2.1 Knowledge Distillation

Knowledge distillation (KD) [33] is the process of transferring knowledge from a teacher model to a student model. It has been successfully used in a wide range of applications, such as machine translation [40], visual recognition [31], speech recognition [10], and recommendation systems [69]. Recent works advanced the field of knowledge distillation by proposing new architectures [77; 80; 1; 55] and objectives [34; 14].

While many KD works study the problem of knowledge transfer within the same modality, cross-modal knowledge distillation [27; 20; 71] tackles the knowledge transfer across different modalities. Gupta et al. [27] transfers the knowledge of a model trained with RGB images to another model for depth maps and optical flow. Do et al. [20] proposes a KD method for visual question answering

[2], where the trilinear (image-question-answer) relational representation of a teacher model is transferred to a bilinear (image-question) student model. Tian et al. [71] combines contrastive learning and knowledge distillation to improve the knowledge transfer between different modalities. Our VIDLANKD transfers the knowledge of a multi-modal teacher model learned from a video dataset to a student language model that tackles natural language understanding tasks.

## 2.2 Language Pretraining

Large-scale pretraining of contextualized language models has seen huge success in natural language processing in recent years. ELMo [57] proposes to pretrain and fine-tune a large recurrent language model, which improves performance on a diverse set of downstream natural language processing tasks. BERT [19] improves the scalability of the pretrain/fine-tune framework by using a transformer [72] language model with a masked language modeling objective. Since then, pretraining of transformer language models has been extensively explored [49; 78; 44; 22; 64; 59; 16] for various natural language understanding [61; 74; 79; 73] and generation tasks [24; 62; 61; 63].

## 2.3 Multi-modal Pretraining

Following the success of language pretraining with transformer models, pretraining of image-text [67; 51; 13; 47; 83; 45] and video-text [66; 54; 85; 53; 46; 70] multi-modal transformers have achieved improvements on numerous multi-modal downstream tasks [2; 76; 84]. The multi-modal transformers take both visual and textual inputs and are pretrained on image-text or video-text pairs with multi-modal masked language modeling objectives. Despite the success on multi-modal downstream tasks, Tan and Bansal [68] finds that the multi-modal pretraining does not improve (and sometimes even harms) the language understanding performance. This is because the scale and diversity of text vocabulary of image-text and video-text datasets are usually smaller than those of text datasets. To utilize the rich vocabulary of text dataset, our VIDLANKD transfers the knowledge of pretrained multi-modal model to a student language model with a large text dataset.

## 2.4 Visually-Grounded Language Learning

A series of works explore using visual information to aid language understanding and generation tasks including co-reference resolution [42; 15], machine translation [81], bilingual lexicon learning [39], and multi-modal contrastive learning [47]. Vokenization [68] proposes the visually-supervised language model, which is closest to our work. Vokenization proposes to supervise a language model to predict a visualized token, called 'voken' for each input text token. Vokens are obtained by a contextualized token-to-image matching model, pretrained on a MS COCO image captioning dataset [12]. In this work, we experiment with alternative objectives which avoid the approximation error from finite voken assignments. In addition, we use HowTo100M [54] video dataset, which provides a more diverse vocabulary as well as richer world commonsense and physical reasoning knowledge.

## 3 Video-Language Knowledge Distillation

### 3.1 Method Overview

We aim to learn a better language representation with the knowledge distilled from visual information. For this, we leverage two kinds of datasets: the aligned muti-modal dataset, $D_{\mathrm{VL}}$: $\{(\mathbf{x}, \mathbf{v})\}$ (e.g., HowTo100M [54]); and the text dataset, $D_{\mathrm{L}}$: $\{\mathbf{x}\}$ (e.g., Wikipedia), where $\mathbf{x}$ is a sentence and $\mathbf{v}$ is a video paired with $\mathbf{x}$. Our knowledge transfer is done in two stages: (1) cross-modal pretraining of a teacher model, $\mathrm{M}^T$, on multi-modal data $D_{\mathrm{VL}}$ (Eq. 1) (2) distilling the knowledge of teacher model to a student model, $\mathrm{M}^S$, on text data $D_{\mathrm{L}}$ (Eq. 2). We illustrate our two-stage knowledge transfer method in Fig. 1.

$$\min_{\theta^T} \mathbb{E}_{\mathbf{x}, \mathbf{v} \sim D_{\mathrm{VL}}} \mathcal{L}^T(\mathrm{M}^T, \mathbf{x}, \mathbf{v}) \tag{1}$$

$$\min_{\theta^S} \mathbb{E}_{\mathbf{x} \sim D_{\mathrm{L}}} \mathcal{L}^{KD}(\mathrm{M}^T, \mathrm{M}^S, \mathbf{x}) \tag{2}$$

Our teacher model $\mathrm{M}^T$ consists a language model $\mathrm{LM}^T$ and a visual encoder $\mathrm{V}^T$. Both $\mathrm{LM}^T$ and $\mathrm{V}^T$ have transformer [72] architectures, where $\mathrm{LM}^T$ takes text tokens $\mathbf{x}$ and $\mathrm{V}^T$ takes video frame

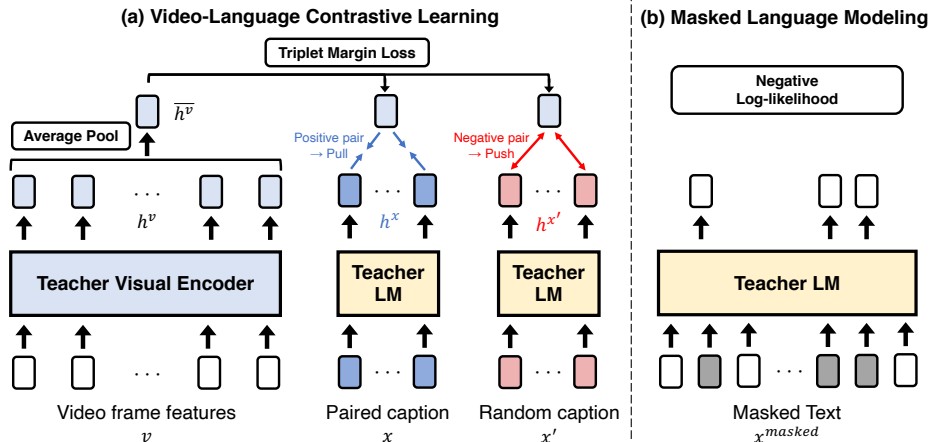

Figure 2: Cross-modal pretraining of our teacher model on a multi-modal dataset (Sec. 3.2). We train our teacher model with (a) video-language contrastive learning and (b) masked language modeling. For video-language contrastive learning, we only illustrate the negative text samples for brevity.

features $\mathbf{v}$ as inputs. Our student model $\mathrm{M}^S$ is a transformer language model $\mathrm{LM}^S$ sharing the same architecture with teacher language model $\mathrm{LM}^T$. As illustrated in Fig. 1(a), we first train teacher models $\mathrm{LM}^T$ and $\mathrm{V}^T$ with contrastive learning and masked language modeling. Then, we distill the knowledge of teacher models to student model $\mathrm{LM}^S$ as in Fig. 1(b). In the following subsections, we discuss the detailed training procedure of teacher (Sec. 3.2, Fig. 2) and student models (Sec. 3.3, Fig. 3).

## 3.2 Teacher Model

We train our teacher model on a multi-modal dataset with two objectives, i.e., video-language contrastive learning (Fig. 2(a)) and masked language modeling (Fig. 2(b)): $\mathcal{L}^T = \mathcal{L}_{CT} + \mathcal{L}_{MLM}$ [2]

**Architecture.** As shown in Figure 2, our teacher model $\mathrm{M}^T$ consists of a language encoder $\mathrm{LM}^T$ and a visual encoder $\mathrm{V}^T$. Both $\mathrm{LM}^T$ and $\mathrm{V}^T$ have similar transformer architecture.[3] For each sentence $\mathbf{x}$, we tokenize it and append a special token [CLS] that represents the entire sentence following Devlin et al. [19]. $\mathrm{LM}^T$ takes $\mathbf{x}$ and outputs contextualized representation $\mathbf{h}^x = \{\mathbf{h}^x_{[\mathrm{CLS}]}, \mathbf{h}^x_1 \cdots \mathbf{h}^x_{|x|}\}$. For each video $\mathbf{v}$, we extract frame-level features $\mathbf{e}^v$ with an off-the-shelf image encoder (see more details in Sec. 4.2). Note that the parameters of the image encoder are not updated to save computation. We feed the frame features $\mathbf{e}^v = \{\mathbf{e}^v_1 \cdots \mathbf{e}^v_{|v|}\}$ to our visual encoder $\mathrm{V}^T$ to get contextualized video frame features $\mathbf{h}^v = \{\mathbf{h}^v_1 \cdots \mathbf{h}^v_{|v|}\}$. We get the final video representation $\overline{\mathbf{h}^v}$ by temporally averaging frame-level features: $\overline{\mathbf{h}^v} = \frac{1}{|v|} \sum_{i=1}^{|v|} \mathbf{h}^v_i$. Different from Tan and Bansal [68], both $\mathrm{LM}^T$ and $\mathrm{V}^T$ parameters are trained from scratch.

**Video-Language Contrastive Learning.** To learn multi-modal grounding, we use a contrastive learning objective that encourages a closer distance between representations of aligned video-text pairs than unaligned pairs, as shown in Fig. 2 (a). For each $\mathbf{x}$, we randomly sample another text $\mathbf{x}'$ from its batch with $\mathbf{x}' \neq \mathbf{x}$. Similarly, for each $\mathbf{v}$, we randomly sample another video $\mathbf{v}'$ from its

---

[2]In our experiments, different weights over the objectives did not significantly change the results.

[3]In our experiments, we use the BERT architecture with two different configurations: 12 layers/768 hidden dimensions ($\mathrm{BERT}_{12L/768H} = \mathrm{BERT}_{BASE}$) and 6 layers/512 hidden dimensions ($\mathrm{BERT}_{6L/512H}$).

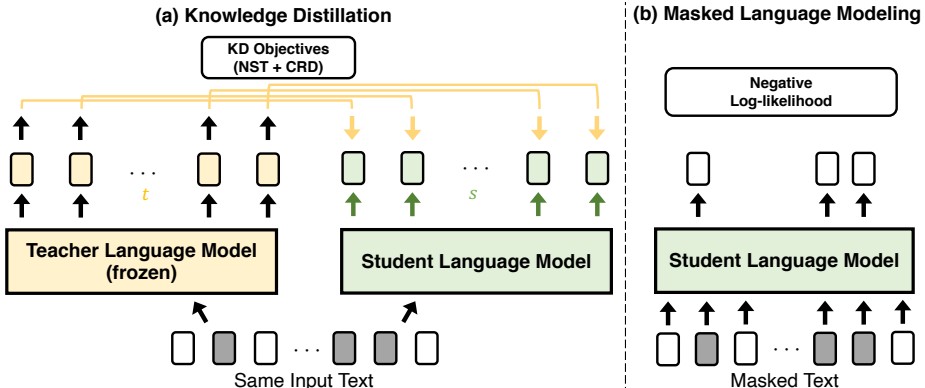

Figure 3: Illustration of our knowledge distillation from teacher language model $\mathrm{LM}^T$ to student language model $\mathrm{LM}^S$ on a text dataset (Sec. 3.3). We train our student model with (a) knowledge distillation objectives and (b) masked language modeling.

batch with $\mathbf{v}' \neq \mathbf{v}$. Then, we use hinge loss $\max\{0, \alpha - pos + neg\}$ on cosine similarities:

$$\mathcal{L}_{CT}(\mathbf{x}, \mathbf{x}', \mathbf{v}, \mathbf{v}') = \sum_i^{|\mathbf{x}|} [\max\{0, \alpha - \cos(\mathbf{h}_i^x, \overline{\mathbf{h}^v}) + \cos(\mathbf{h}_i^{x'}, \overline{\mathbf{h}^v})\} \tag{3}$$
$$+ \max\{0, \alpha - \cos(\mathbf{h}_i^x, \overline{\mathbf{h}^v}) + \cos(\mathbf{h}_i^x, \overline{\mathbf{h}^{v'}})\}]$$

where $\alpha$ is the margin between the similarities of a positive pair and a negative pair. Different from previous methods [7; 32] that exploit sentence-level contrastive loss, we follow [68] to construct a token-level contrastive loss (triplet margin loss) that grounds the visual information to each contextualized token output. This fine-grained contrastive loss will help the token-level knowledge distillation in Sec. 3.4.

**Masked Language Modeling.** For better language understanding in our teacher model, we follow BERT [19] to use masked language modeling (MLM) objective (Fig. 2(b)). By replacing 15% of tokens in $\mathbf{x}$ with a special token [MASK], we obtain a masked text $\mathbf{x}^{\mathrm{masked}}$ with the same length. The model takes $\mathbf{x}^{\mathrm{masked}}$ as input and learns to predict the tokens by minimizing the negative log-likelihoods: $\mathcal{L}_{\mathrm{MLM}}(\mathbf{x}, \mathbf{x}^{\mathrm{masked}}) = -\sum_{i \in \mathrm{Mask}} \log p(\mathbf{x}_i \mid \mathbf{x}^{\mathrm{masked}})$, where $\mathrm{Mask}$ refers to the indices of masked tokens.

## 3.3 Student Model

After we train a teacher model on a multi-modal dataset, we transfer its knowledge to a student model on a text dataset. Following Kim and Rush [40], we train our student model with a sum of masked language modeling and two knowledge distillation objectives, NST and CRD (see Sec. 3.4):

$$\mathcal{L}^{\mathcal{S}} = \mathcal{L}_{\mathrm{MLM}} + \mathcal{L}_{\mathrm{NST}}^{\mathrm{KD}} + \mathcal{L}_{\mathrm{CRD}}^{\mathrm{KD}} \tag{4}$$

**Architecture.** As shown in Fig. 3 (a), our student model $\mathrm{M}^S$ is a language model $\mathrm{LM}^S$ with the same transformer architecture as the teacher language model $\mathrm{LM}^T$. We train $\mathrm{LM}^S$ from scratch. Following previous works [11; 26], we introduce a multi-layer perceptron (MLP) distillation head on top of the last hidden states of $\mathrm{LM}^S$. In our ablation study in appendix, we find that adding a distillation head slightly improves the distillation performance.

## 3.4 Knowledge Distillation Objectives

We next describe the knowledge distillation (KD) objectives used in transferring knowledge from the teacher model $\mathrm{LM}^T$ (Sec. 3.2) to this student model $\mathrm{LM}^S$. Note that the weights of the teacher model $\mathrm{LM}^T$ are frozen during the knowledge distillation process since the teacher model should not be affected by the student model's performance. Following Kim and Rush [40], we use the knowledge

distillation objective combined with the MLM objective (Fig. 3 (b)). Concretely, we use the same input text mask for MLM and KD objectives. While we calculate the MLM loss only on masked positions, we calculate KD losses using all hidden states following Clark et al. [16] (Fig. 3).

We study the following KD objectives: Soft-label [33], L2 Regression [3], Neuron Selectivity Transfer (NST) [34], Contrastive Representation Distillation (CRD) [71], and Vokenization [68]. In our experiments comparing different KD objectives (Table 5), NST and CRD perform best, while the combination of them improved the performance even further. Therefore, we propose NST+CRD for our cross-modal knowledge distillation objective.

**Soft Label:** Hinton et al. [33] proposed a knowledge transfer method by taking a teacher model prediction with temperature scaling as a 'soft label'. We minimize cross-entropy between $P^T(y|x)$ and $P^S(y|x)$, i.e., the word output probabilities of $\mathrm{LM}^T$ and $\mathrm{LM}^S$ given the input text $\mathbf{x}$ respectively:

$$\mathcal{L}_{\text{soft-label}}^{\text{KD}}(\mathbf{x}) = -\sum_{i=1}^{|\mathbf{x}|} \sum_{z \in Z} P^T(y_i = z|\mathbf{x}) \log P^S(y_i = z|\mathbf{x}) \tag{5}$$

where $Z$ is the word vocabulary. Following Hinton et al. [33], we divide the softmax logits of $\mathrm{LM}^T$ and $\mathrm{LM}^S$ by a temperature parameter $\tau = 2.0$. Note that for soft-label KD, we reuse the LM head, instead of learning an additional distillation head.

**L2 Regression**: Following Ba and Caruana [3] which uses feature regression for KD, we minimize the squared L2 distance between $\mathbf{s}(\mathbf{x})$ and $\mathbf{t}(\mathbf{x})$, the last hidden states of $\mathrm{LM}^T$ and $\mathrm{LM}^S$ given input text $\mathbf{x}$:

$$\mathcal{L}_{\text{Regression}}^{\text{KD}}(\mathbf{x}) = \sum_{i=1}^{|\mathbf{x}|} \|\mathbf{s}(\mathbf{x})_i - \mathbf{t}(\mathbf{x})_i\|_2^2 \tag{6}$$

**Neuron Selectivity Transfer (NST)**: NST [34] is a KD method that transfers heatmap like spatial activation patterns of teacher neurons to student neurons. We transfer the sequential activation patterns of $\mathbf{t}(\mathbf{x}) \in \mathbb{R}^{|\mathbf{x}| \times d}$ to $\mathbf{s}(\mathbf{x}) \in \mathbb{R}^{|\mathbf{x}| \times d}$, where $\mathbf{t}(\mathbf{x})$ and $\mathbf{s}(\mathbf{x})$ are the last hidden states of $\mathrm{LM}^T$ and $\mathrm{LM}^S$ given input text $\mathbf{x}$, and $d$ is the hidden state dimension (# neurons). Following Huang and Wang [34], we use the squared maximum mean discrepancy (MMD) [25] with kernel trick to measure the distance between the activation patterns of student neurons $\{\mathbf{s}(\mathbf{x})_{*,i}\}_{i=1}^d$ and teacher neurons $\{\mathbf{t}(\mathbf{x})_{*,j}\}_{j=1}^d$:

$$\begin{aligned}
\mathrm{MMD}^2(\mathbf{x}) = &\frac{1}{d^2} \sum_{i=1}^d \sum_{i'=1}^d k\left[\mathbf{s}(\mathbf{x})_{*,i}; \mathbf{s}(\mathbf{x})_{*,i'}\right] + \frac{1}{d^2} \sum_{j=1}^d \sum_{j'=1}^d k\left[\mathbf{t}(\mathbf{x})_{*,j}; \mathbf{t}(\mathbf{x})_{*,j'}\right] \\
&- \frac{2}{d^2} \sum_{i=1}^d \sum_{j=1}^d k\left[\mathbf{s}(\mathbf{x})_{*,i}; \mathbf{t}(\mathbf{x})_{*,j}\right]
\end{aligned} \tag{7}$$

where we use Gaussian kernel $k[\mathbf{s}; \mathbf{t}] = \exp\left(-\frac{\|\mathbf{s}-\mathbf{t}\|_2^2}{2\sigma^2}\right)$ with $\sigma = 1$. We transfer the teacher activation patterns to the student by minimizing squared MMD: $\mathcal{L}_{\text{NST}}^{\text{KD}}(\mathbf{x}) = \mathrm{MMD}^2(\mathbf{x})$

**Contrastive Representation Distillation (CRD)**: CRD [71] is a KD objective which maximizes the mutual information between the teacher and student representations with contrastive learning. Let's denote $\mathbf{s} \in S$ and $\mathbf{t} \in T$ as student and teacher representations given $\mathbf{x}$. We are given 1 positive pair (drawn from the joint distribution) for every $N$ (batch size) negative pairs (drawn from the product of marginals; independent randomly drawn inputs from $T$ and $S$). Following [71], we maximize the lower bound of mutual information between $\mathbf{s}$ and $\mathbf{t}$ by minimizing the following term:

$$\mathcal{L}_{\text{CRD}}^{\text{KD}}(\mathbf{x}) = -\mathbb{E}_{q(\mathbf{s}, \mathbf{t}|\text{postive})}[\log h(\mathbf{s}, \mathbf{t})] - N \cdot \mathbb{E}_{q(\mathbf{s}, \mathbf{t}|\text{negative})}[\log(1 - h(\mathbf{s}, \mathbf{t}))] \tag{8}$$

$$h(\mathbf{s}, \mathbf{t}) = \frac{\exp\left(f_1(\mathbf{s})^\top f_2(\mathbf{t})\right)}{\exp\left(f_1(\mathbf{s})^\top f_2(\mathbf{t})\right) + \frac{N}{M}}$$

where $M$ is the cardinality of the dataset, $f_1, f_2$ are learned linear layers followed by $L2$ normalization, which map the student and teacher representations into a same feature space. Since a large $N$ leads

to a tight mutual information lower bound, following [71], we implement a memory buffer that stores the latent features of each data sample computed from previous batches. Therefore, during training we can efficiently retrieve a large number of negative samples from the memory buffer. Note that since CRD is based on contrastive learning, it is the only KD objective where student and teacher language models can take different inputs.

**Vokenization**: Vokenization [68] could be viewed as a knowledge distillation method, where token-level text-to-image retrieval results (called 'vokens') of a multi-modal matching model are used as labels for a student language model. For the i-th input token $\mathbf{x}_i$, we calculate cosine similarity between the i-th teacher language model hidden state $\mathbf{t}(\mathbf{x})_i$ and a video feature $\mathbf{v}$. Out of 30K pre-selected videos, we select a video that maximizes cosine similarity and use it as the voken for $\mathbf{x}_i$. By denoting the voken of $\mathbf{x}_i$ as $\mathbf{voken}_i$, we formulate our vokenization-based KD objetive as:

$$\mathcal{L}_{\text{Voken}}^{\text{KD}}(\mathbf{x}) = -\sum_{i=1}^{|\mathbf{x}|} \log P_{\text{voken}}^{S}(y_i = \mathbf{voken}_i | \mathbf{x}) \tag{9}$$

where $P_{\text{voken}}^{S}(y|\mathbf{x})$ is the voken classification probabilities of $\text{LM}^{S}$ given input text $\mathbf{x}$. We experiment with vokenization-based KD by retrieving vokens from images and videos (see Table 6 of Sec. 5.2). Note that vokenization suffers from approximation error; it's hard to cover diverse textual concepts with 30K vokens. This motivates us to experiment with different 'soft' KD objectives described in this section (see Table 5 of Sec. 5.2).

## 4 Experimental Setup

### 4.1 Datasets

**Video-Text Dataset.** We use HowTo100M [54] for cross-modal pretraining of our teacher model (Sec. 3.2). HowTo100M has 1.22M videos totaling 136M video clips with total duration of 134,472 hours describing over 23K different visual tasks. There are 138M captions, 568M tokens with 633K distinct tokens.

**Text Pretraining Dataset.** To transfer the knowledge from our teacher language models to student language models (Sec. 3.3), we follow Tan and Bansal [68] to use English Wikipedia. For ablation studies (Sec. 5.2), we use Wiki103 [52], a widely used subset of English Wikipedia. There are 2.9B tokens and 120M sentences in English Wikipedia, and 111M tokens and 4.2M sentences in Wiki103.

**Text Downstream Dataset.** Following Tan and Bansal [68], we finetune our models on GLUE [73], SQuAD [61] 1.0 and SQuAD2.0 [60], and SWAG [79] to assess the pretraining performance. Since some smaller tasks in GLUE are reported as unstable in recent papers [21], we evaluate on the four largest datasets of GLUE: SST-2 [9], QNLI [61], QQP [35], and MNLI [74]. In addition, we also evaluate our models on the GLUE diagnostics [73], PIQA [6], and TRACIE [82] to measure its linguistic knowledge, physical reasoning, and temporal reasoning abilities.

### 4.2 Video Feature Representations

Following Miech et al. [54], we encode video features by concatenating features from a 2D frame-level image encoder and a 3D video encoder in channel dimension. Note that the parameters for 2D image encoder and 3D video encoder are not updated.

For the 2D image encoder, we sample video frames by 1fps (frame/second). The 2D image encoder outputs features for each frame individually. We experiment with ResNet-152 [30] pretrained on ImageNet-1K [18] and CLIP [58] image encoder (ViT-B/32 [23]). In contrast to conventional image encoders trained with image label classification, the CLIP image encoder is trained to match a corresponding natural language description by large-scale contrastive learning. We discuss if this natural language supervision can help our cross-modal KD in Sec. 5.1.

For the 3D video encoder, we use 3D-ResNeXt-152[4] [75; 28; 36] trained from a combination of publicly available datasets: ActivityNet [8], Kinetics [37], UCF-101 [65], and HMDB-51 [43]. The 3D video encoder processes 24fps videos with 3D convolution and yields features at 1.5fps. Then we sub-sample the features to 1fps to match the frame rate of 2D image encoder.

---

[4] https://github.com/kenshohara/3D-ResNets-PyTorch

Table 1: Cross-modal knowledge distillation results of BERT$_{12L/768H}$ student language model on 7 downstream NLU tasks. In the first block, we include the image-based vokenization (Img-Voken) and its text-only pretrained baseline performance from Tan and Bansal [68]. In the second block, we compare our cross-modal KD method (NST+CRD) to video-based vokenization (Vid-Voken) and a text-only pretrained baseline. [†]EM refers to 'Exact Match'.

| | SST-2 Acc | QNLI Acc | QQP Acc | MNLI Acc | SQuAD v1.1 EM[†] | SQuAD v2.0 EM | SWAG Acc | Avg. |
|---|---|---|---|---|---|---|---|---|
| BERT$_{12L/768H}$ [68] | 89.3 | 87.9 | 83.2 | 79.4 | 77.0 | 67.7 | 65.7 | 78.6 |
| + KD (Img-Voken) [68] | 92.2 | 88.6 | 88.6 | 82.6 | 78.8 | 68.1 | 70.6 | 81.4 |
| BERT$_{12L/768H}$ | 89.0 | 88.0 | 86.2 | 79.2 | 77.2 | 68.0 | 65.0 | 78.9 |
| + KD (Vid-Voken) w/ ResNet | 93.4 | 89.2 | 88.7 | 83.0 | 78.9 | 68.7 | 70.0 | 81.7 |
| + KD (Vid-Voken) w/ CLIP | 94.1 | **89.8** | 89.0 | 83.9 | 79.2 | 68.6 | 71.6 | 82.3 |
| + KD (NST+CRD) w/ ResNet | 94.2 | 89.3 | 89.7 | 84.0 | 79.0 | **68.9** | 71.8 | 82.4 |
| + KD (NST+CRD) w/ CLIP | **94.5** | 89.6 | **89.8** | **84.2** | **79.6** | 68.7 | **72.0** | **82.6** |

## 4.3 Implementation Details

For the student distillation head, we use a two-layer MLP with ReLU activation. For both student and teacher language models, following previous works [49; 17; 68], we truncate input text that is longer than 128 tokens. We truncate videos features that are longer than 512 frames. We use an AdamW [41] optimizer with learning rate 2e-4 and weight decay [50] of 0.01. We reserve 10K samples of the HowTo100M dataset as validation data. We train the teacher model until it converges on validation data. For downstream tasks, we report the results on the validation sets. We train 3 epochs with a learning rate of 1e-4 and a batch-size of 32 for all downstream tasks. We use hinge loss margin $\alpha = 1.0$ for $\mathcal{L}_{CT}$ (Eq. 3). We implement our models with PyTorch 1.5 [56] and train them with Nvidia GeForce RTX 2080ti GPUs. For teacher pretraining, we use 4 GPUs for BERT$_{12L/768H}$ and BERT$_{6L/512H}$ models for 7 days and 2.5 days respectively. For knowledge distillation, we use 4 GPUs for BERT$_{12L/768H}$ and BERT$_{6L/512H}$ models for 10 days and 3 days respectively.

# 5 Results and Analysis

## 5.1 Primary Downstream Task Results

In the first block of Table 1, we include the image-based vokenization (Img-Voken) and their text-only pretrained baseline from Tan and Bansal [68].[5] Given our reproduced text-only baseline shows a similar average performance (78.9 vs 78.6), our student models distilled from NST+CRD are much better (82.6 vs 81.4). We discuss the comparison between video-based and image-based KD in detail in the following ablation study in comparison to vokenization (Table 6).

In the second block of Table 1, we compare our proposed cross-modal KD method (NST+CRD) to video-based vokenization (Vid-Voken) and a non-KD baseline (BERT$_{12L/768H}$) which is only pretrained on text. We can see both cross-modal KD methods (i.e., KD and Vid-Voken) significantly outperform the text-only baseline across all 7 downstream tasks. We also experiment with different 2D frame encoders (Sec. 4.2): ResNet and CLIP. For both Vid-Voken and NST+CRD, we observe CLIP further improves the performance results over ResNet, indicating using a strong visual encoder helps the teacher training and thus benefits the knowledge distillation.

## 5.2 Ablation Studies

In this section, we conduct a comprehensive ablation study to show the effectiveness of our proposed methods. For all ablation experiments, we use BERT$_{6L/512H}$ architecture for student and teacher language models. We use ResNet-152 for 2D frame encoder and 3D-ResNeXt-152 for 3D frame encoder (Sec. 4.2). Wiki103 [52] is used for student model training. We also perform ablation experiments on the effect of additional distillation head in appendix.

---

[5]Vokenization uses pretrained BERT checkpoint for its 'teacher' (vokenizer) model but we train our teacher language model fully from scratch.

**Text-only Pretraining.** Our cross-modal KD improves the performance on downstream NLU tasks significantly (Sec. 5.1). Where does the improvement come from, video or text? To answer this question, we conduct text-only pretraining of BERT$_{6L/512H}$ on Wiki103 text (111M tokens), HowTo100M captions (568M tokens) and compare them to a no-pretrain baseline. In Table 2, while both pretrained models improve the performance over the no-pretrain baseline, Wiki103-trained model outperforms HowTo100M-trained model (which has more tokens) significantly. This indicates that our KD methods improve NLU performance because of multimodal grounding, instead of just the larger corpus.

Table 2: Text-only pretraining results of BERT$_{6L/512H}$ pretrained on Wiki103, HowTo100M captions, and no-pretrain baseline.

| Pretrained on | SST-2 | QNLI | QQP | MNLI |
|---|---|---|---|---|
| No-Pretrain | 79.6 | 61.5 | 72.7 | 61.6 |
| Wiki103 (Formal language) | **88.8** | **84.9** | **85.3** | **77.4** |
| HowTo100M (ASR captions) | 83.3 | 78.5 | 83.7 | 71.5 |

**Effect of Teacher Training Objectives.** We here analyze the teacher training objectives by comparing the corresponding distilled student model results. In Table 3, the teacher model trained solely with MLM (+KD from $T^{MLM}$) does not significantly change the student model performance. At the same time, the teacher model trained with only visual supervision, i.e., contrastive objective (+KD from $T^{CT}$), improves the result. This illustrates the motivation to perform knowledge transfer from a visually supervised MLM model. Lastly, combining the MLM and the contrastive objective (+KD from $T^{MLM+CT}$) in teacher model training shows the best student results.

Table 3: Ablation results showing the effect of the teacher model's training objectives. NST is used for knowledge distillation.

| | SST-2 | QNLI | QQP | MNLI |
|---|---|---|---|---|
| BERT$_{6L/512H}$ | 88.8 | 84.9 | 85.3 | 77.4 |
| +KD from $T^{MLM}$ | 88.1 | 83.1 | 85.6 | 77.4 |
| +KD from $T^{CT}$ | 88.9 | **85.2** | 86.2 | 77.5 |
| +KD from $T^{MLM+CT}$ | **91.1** | 85.0 | **87.4** | **78.4** |

**Two-stage PT vs. Cross-modal KD.** In Table 4, we compare two-stage pretraining with a single model to our proposed cross-modal KD approach. For single model baselines, we use text-only (MLM on Wiki103), video-only (MLM+CT on HowTo100M), and two-stage (video-then-text) pretraining. While the two-stage pretraining shows better results than the text/video-only pretraining, our VIDLANKD outperforms all baselines on GLUE tasks, especially on SST-2 and MNLI.

Table 4: Comparison of pretraining on text, video, both (Two-stage PT), and our VidLanKD.

| Model | SST-2 | QNLI | QQP | MNLI |
|---|---|---|---|---|
| Text PT | 88.8 | 84.9 | 85.3 | 77.4 |
| Video PT | 84.0 | 78.9 | 84.2 | 73.1 |
| Two-Stage PT | 90.3 | **85.0** | 87.2 | 76.9 |
| VIDLANKD | **91.1** | 85.0 | **87.4** | **78.4** |

**KD Objectives Comparison.** In Table 5, we compare different knowledge distillation objectives introduced in Sec. 3.4. The student models trained with NST [34] and CRD [71] show the best finetuning performance on downstream tasks. When combining NST and CRD, performance further improves with marginal additional computation cost, hence we propose to use NST+CRD for our cross-modal knowledge distillation.

Table 5: Ablation of knowledge distillation objectives.

| | SST-2 | QNLI | QQP | MNLI |
|---|---|---|---|---|
| BERT$_{6L/512H}$ | 88.8 | 84.9 | 85.3 | 77.4 |
| +KD-Soft label | 87.2 | 84.4 | 86.4 | 76.6 |
| +KD-Regression | 88.8 | 84.8 | 87.1 | 78.1 |
| +KD-Vid Voken | 89.7 | 85.5 | 86.5 | 77.8 |
| +KD-NST | **91.1** | 85.0 | **87.4** | **78.4** |
| +KD-CRD | 90.0 | **85.5** | 87.3 | 78.3 |
| +KD-NST+CRD | **91.5** | **85.8** | **87.4** | **78.7** |

**Comparison to Vokenization.** In Table 6, we compare NST [34] and Vokenization [68] in both image and video-level teacher model supervision. For video-level supervision, we provide our visual encoder with the whole video features (Sec. 4.2). For image-level supervision, we provide our visual encoder only with 2D features of the middle frame for each video clip. With image-level super-

Table 6: Comparison between vokenization (Voken) and NST with image and video-level supervision.

| | SST-2 | QNLI | QQP | MNLI |
|---|---|---|---|---|
| BERT$_{6L/512H}$ | 88.8 | 84.9 | 85.3 | 77.4 |
| +KD-Voken (Image) | 89.3 | 84.4 | 86.0 | 77.5 |
| +KD-NST (Image) | 88.9 | 85.0 | 86.3 | 77.2 |
| +KD-Voken (Video) | 89.7 | **85.5** | 86.5 | 77.8 |
| +KD-NST (Video) | **91.1** | 85.0 | **87.4** | **78.4** |

vision (first block), Vokenization and NST show comparable performance. However, with video-level supervision (second block), NST outperforms Vokenization on 3 out of 4 tasks. The gap in the video domain might come from voken approximation error, where each image or video input is approximated with one of 30K predefined vokens. Since videos usually contain more diverse contents than images, the voken approximation error would be amplified in video-level supervision, whereas our NST distillation avoids this issue.

## 5.3 Analyzing the Knowledge Learned from Video

In this subsection, we analyze the knowledge that our language models learn from video via cross-modal knowledge distillation. To measure linguistic knowledge and physical/temporal reasoning ability, we show results of our models on the GLUE diagnostics [73], the Physical Interaction Question Answering (PIQA) [6], and TRACIE [82]. In addition, we visualize the learned multi-modal grounding ability of our model with text-to-video retrieval.

Table 7: Finetuning performance on GLUE diagnostics [73], PIQA [6] and TRACIE [82] datasets, which measure the linguistic knowledge, physical and temporal reasoning capabilities of language models, respectively.

| | GLUE diagnostics | | | | PIQA | TRACIE |
|---|---|---|---|---|---|---|
| | Lexicon | Predicate | Logic | Knowledge | | |
| $BERT_{6L/512H}$ | 53.0 | 64.2 | 44.5 | 44.0 | 56.9 | 63.4 |
| + KD-NST | 53.3 (+0.3) | 63.7 (-0.5) | 44.8 (+0.3) | 48.6 (**+4.6**) | 60.0 (**+3.1**) | 66.7 (**+3.3**) |

**Linguistic Knowledge.** GLUE diagnostics dataset [73] evaluates sentence understanding through natural language inference (NLI) problems. The dataset consists of sentence pairs labeled with their entailment relations (entailment, contradiction, or neutral) in both directions and tagged with a set of entailment labels. Each example in the dataset is labeled with 4 categories of linguistic phenomena: (1) lexical semantics, (2) predicate-argument structure, (3) logic, and (4) knowledge (including common sense). In Table 7, we compare the baseline language model ($BERT_{6L/512H}$ pretrained on Wiki103) to our NST-distilled model. We finetune the models on MNLI [74] that has the same format and test on GLUE diagnostics. We observe a large gain on the knowledge category (which involves common sense and external world knowledge) while there are no significant differences on other categories. This suggests that our student model learns the external, grounded world knowledge in the teacher model and the video-text dataset.

**Physical and Temporal Reasoning.** PIQA [6] is a question answering dataset evaluating physical interactions and commonsense reasoning. TRACIE [82] is a temporal reasoning benchmark on implicit events, which are not mentioned explicitly in natural language text but can be inferred from it. In Table 7, our $BERT_{6L/512H}$ distilled with NST significantly outperform the text-only pretrained baseline on both benchmarks. The finding suggests (consistent with the GLUE diagnostics findings above) that video knowledge distillation also helps improve the physical and temporal reasoning capabilities of the language model. See appendix for the more detailed discussion on the PIQA and TRACIE experiment.

## 6 Conclusion

We introduce VIDLANKD, a novel cross-modal knowledge distillation method to help general language understanding. Our teacher model is first trained on a video-text dataset, and then we transfer its knowledge to a student language model with a text dataset. Via the distillation objectives and video-text datasets, our method overcomes the limitations of the recent vokenization method. We empirically demonstrate that our VIDLANKD improves on several NLU tasks over models trained by pure-text or vokenization. We conduct comprehensive ablation analysis to show the effectiveness of each proposed component. We also illustrate the linguistic knowledge and physical/temporal commonsense reasoning learned from videos, and visualize our model's multi-modal grounding ability.

# Acknowledgments

We thank the reviewers for their helpful comments. We thank Yixin Nie and Gabriel Ilharco for useful dataset suggestions. This work was supported by ARO-YIP Award W911NF-18-1-0336, DARPA MCS Grant N66001-19-2-4031, DARPA KAIROS Grant FA8750-19-2-1004, Google Focused Research Award, and Bloomberg Data Science Ph.D. Fellowship. The views, opinions, and/or findings contained in this article are those of the authors and not of the funding agency.

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
