# Supplementary Materials for
# VIDLANKD: Improving Language Understanding via Video-Distilled Knowledge Transfer

**Zineng Tang**    **Jaemin Cho**    **Hao Tan**    **Mohit Bansal**
UNC Chapel Hill
{terran, jmincho, haotan, mbansal}@cs.unc.edu

In this appendix, we start with describing the experimental setup details (Sec. A). We provide ablation study on distillation head (Sec. B), details of physical (Sec. C) and temporal (Sec. D) reasoning analysis, details of text-to-video visualization (Sec. E), broader impacts and limitations (Sec. F), licenses for code/model/dataset (Sec. G), and NeurIPS Paper Checklist (Sec. H).

## A    Experimental Setup

**Video Voken Sampling.**    To ensure the diversity of video vokens, we first select a video for each of 23K visual task. For the remaining 7K vokens, we randomly select 7K visual tasks, then select a video from each visual task. Each sampled video from 30K has on average around 100 clips. We select one clip from each video with length ranging from 1 to 20 seconds.

## B    Additional Distillation Head

To investigate whether the additional MLP distillation head (Sec.3.3 in the main paper) affects the distillation performance, we do an ablation by conducting knowledge distillation directly on the last hidden states of student language models. As we see in Table 1, for both NST and CRD, the performance drops on all downstream tasks when distillation heads are removed. This finding is consistent with recent works [3; 4].

Table 1: Ablation results of additional distillation heads for student language models.

|  | SST-2 | QNLI | QQP | MNLI |
|---|---|---|---|---|
| BERT$_{6L/512H}$ | 88.8 | 84.9 | 85.3 | 77.4 |
| +KD-NST | **91.1** | 85.0 | **87.4** | **78.4** |
| +KD-CRD | 90.0 | **85.5** | 87.3 | 78.3 |
| +KD-NST (w/o head) | 89.4 (-0.7) | 84.8 (-0.2) | 86.7 (-0.7) | 77.0 (-1.4) |
| +KD-CRD (w/o head) | 88.9 (-0.1) | 85.1 (-0.4) | 86.6 (-0.7) | 77.8 (-0.5) |

## C    Physical Reasoning Details

PIQA [1] is a physical commonsense reasoning dataset with a format of choosing an answer among two hypotheses given context. In Table 2, we compare the accuracy of text-only pretraining, image-based KD and video-based KD on PIQA. While the image-based KD helps to improve accuracy from text-only pretrained model, our VIDLANKD further improves the results. In Tables 3 and 4, we

Table 2: Performance on PIQA with teacher trained with images or video supervision. NST is used as KD objective.

| | TRACIE Accuracy |
|---|---|
| $\text{BERT}_{6L/512H}$ | 56.9 |
| + Image KD | 58.9 |
| + VIDLANKD | **60.0** |

Table 3: PIQA test set examples comparing text-only vs. video grounding. GT stands for ground-truth labels. Text-only refers to the text-only baseline ($\text{BERT}_{6L/512H}$). Ours refers to VIDLANKD student model distilled with NST objective from video supervised teacher model.

| Context | Hypothesis 1 | Hypothesis 2 | GT | Text-only | Ours |
|---|---|---|---|---|---|
| 1. to remove a screw from a board, | (a) place the tip of the screwdriver into the top of the screw and twist in a clockwise direction. | (b) place the tip of the screwdriver into the top of the screw and twist in a counter clockwise direction. | (b) | (a) | (b) |
| 2. how to grow a plant. | (a) bury seed in sand and add 1 cup of water daily. | (b) bury seed in soil and add 1 cup of water daily. | (b) | (a) | (b) |

provide PIQA question examples and related video clips from HowTo100M that could help models to answer the questions.

**Visual Grounding Improves Physical Reasoning.** In Table 3, the first video clip (`https://www.youtube.com/watch?v=ASjB-GtyIZE`) illustrates how to fix a car cup holder that involves removing a screw with a screwdriver, which helps models to learn the action of how to remove a screw from another object. From the second video clip (`https://www.youtube.com/watch?v=NQCuOKFwQ4Q`), the model can learn from the visual of planting in soil, which helps models to identify the correct action on planting.

**Video vs. Image Grounding.** Videos can convey more temporal information such as actions/motions. Video captions (e.g., HowTo100M) also have a larger vocabulary coverage than image captions (e.g., CC or SBU) thus more words could be effectively grounded. Therefore, videos can provide richer visual information than images. In Table 4, The first video clip (`https://www.youtube.com/watch?v=38FqlXKZ6LA`) illustrates how to cut wood with a band saw, which helps models to answer the question. The second video clip (`https://www.youtube.com/watch?v=MMtiszBnpuc`)

Table 4: PIQA test set examples comparing video vs. image grounding. GT stands for ground-truth labels. Image KD refers to our student model distilled with NST objective from image-supervised teacher model. Ours refers to VIDLANKD student model distilled with NST objective from video-supervised teacher model.

| Context | Hypothesis 1 | Hypothesis 2 | GT | Image KD | Ours |
|---|---|---|---|---|---|
| 1. how to cut wood on a band saw. | (a) get the piece of wood you want to cut and put on your safety equipment. start the saw and cut. | (b) start the band saw and put your wood on the top. push it through the blade and let it drop to the floor. | (a) | (b) | (a) |
| 2. how do you properly prepare a steak. | (a) take the steak out of warm storage and let come to room temperature, generously add salt and pepper to both sides and let sit for 10 minutes. | (b) take the steak out of cold storage and let come to room temperature, generously add salt and pepper to both sides and let sit for 10 minutes. | (b) | (a) | (b) |

Table 5: TRACIE test set examples. Ent. and Con. stand for Entailment and Contradiction, respectively. GT stands for ground-truth labels. Text-only refers to the text-only baseline ($\text{BERT}_{6L/512H}$). Ours refers to our student model distilled with NST objective (+KD-NST).

| Context (Premise) | Hypothesis | GT | Text-only | Ours |
|---|---|---|---|---|
| "One day, Ernie went on a walk in the park." Ernie walked by the tennis courts and saw two beautiful women playing. "He had never played tennis before, but he decided to learn." "The next day he went to the park, and the ladies were there again." "They invited him to join them, and eventually one became his wife." | Ernie bought himself a tennis racquet **ends after** the next day he went back to the park. | Con. | Con. | Con. |
| Tim was visiting his grandparents. They didn't have wifi or fast internet. Their connection was still using dial up. Tim tried to use the internet but it was just too slow. He decided to just use his smart phone instead. | Dial up internet is not as good **starts before** Tim visit his grandparents | Ent. | Con. | Ent. |
| Paul hates his job. Everyday at work he gets angry and says mean things to people. Paul's boss gave him a verbal warning about his attitude at work. Currently Paul is on a performance plan at work. Next month Paul will be fired. | Paul is not friendly. **starts after** Paul hat his job | Ent. | Con. | Ent. |

illustrates a brisket recipe where beef is marinated and stored in a 'cold' fridge, which helps our model to answer the question.

## D  Temporal Reasoning Details

As described in Sec.5.3 in the main paper, to measure the temporal understanding ability learned from our video-text pretraining, we fine-tune our model on TRACIE [5], a temporal reasoning benchmark on implicit events — events that are not mentioned explicitly in natural language text but can be inferred from it. We provide three examples from TRACIE test set in Table 5. As illustrated in the table, TRACIE is a textual entailment task where a model infers whether a hypothesis containing a temporal comparator $\in \{\texttt{starts}, \texttt{ends}\}$ and a relation $\in \{\texttt{before}, \texttt{after}\}$ corresponds to a premise. Following [5], we use the *uniform-prior* training setting which removes the statistical correlation between comparators and relations. Table 6 shows the student language model distilled with our VIDLANKD (+KD-NST) outperforms the accuracy of the text-only baseline ($\text{BERT}_{6L/512H}$) by 3.3%. In the right three columns of Table 5, we show the ground truth labels and model predictions for three examples. While our student model correctly predicts all three examples, the text-only baseline fails in the last two examples. We conjecture that it is hard to understand the meaning of words that require temporal understanding, such as 'before' and 'after', only from text. HowTo100M videos consist of multiple events with corresponding ASR captions, which could help models to learn the temporal relations.

Table 6: Performance on TRACIE *uniform-prior* training setting.

| | TRACIE Accuracy |
|---|---|
| $\text{BERT}_{6L/512H}$ | 63.4 |
| +KD-NST | **66.7** |

## E  Visualization: Text-to-Video Retrieval

Our teacher language model learns to predict a corresponding video feature for each input text token (Sec. 3.2) , and our student language model tries to follow the teacher's prediction. To visualize the learned multi-modal grounding, we experiment with text-to-video retrieval using our teacher and student language models. In Fig. 1 and 2 we provide the top 3 text-to-video retrieval results from

Query: "The expansion of agriculture, commerce, trade, and transportation between civilizations in different regions offered cooks many new ingredients."

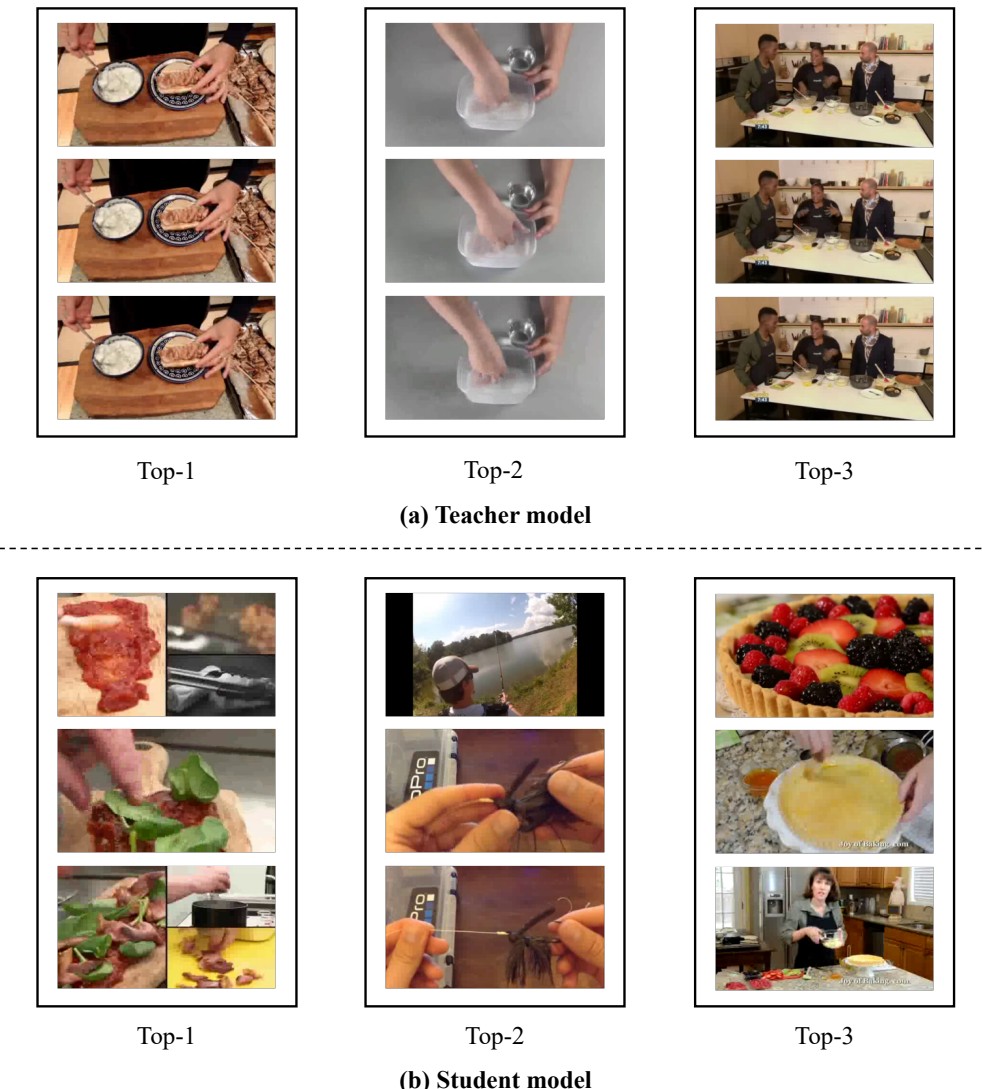

(a) Teacher model

(b) Student model

Figure 1: Text-to-video retrieval results from our teacher and student language model.

teacher and student models using same input sentences. We observe that, in many cases, both our teacher and student model can retrieve video clips that are semantically aligned to input text. Note that this is a surprising and positive result because our student model does not see any visual input during its training (Sec. 3.3), which means the multi-modal grounding ability is learned solely from the knowledge distillation on text dataset.

We use BERT$_{6L/512H}$ architecture for both teacher and student (KD-NST+CRD) language models. We sample sentences from Wikipedia and conduct text-to-video retrieval on the 60K video clips sampled from HowTo100M. For sentence feature, we use the average of the last hidden states of language models. Then we calculate the cosine similarity between the video and sentence features for relevance score.

Query: "As an outcome of these changes, craftspeople today increasingly make use of semi-finished components or materials and adapt these to their customers' requirements or demands."

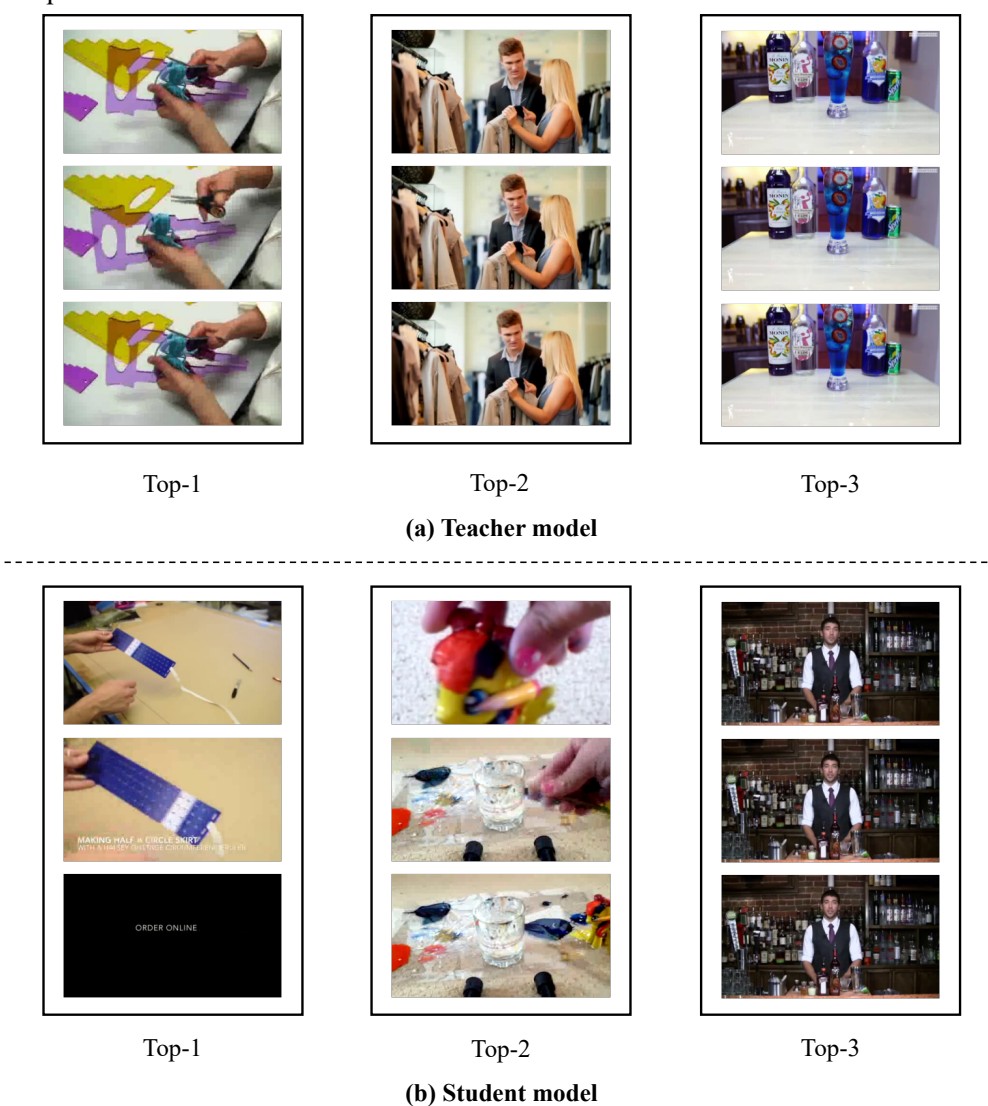

Figure 2: More text-to-video retrieval results from our teacher and student language model.

## F  Broader Impacts and Limitations

There are some risks with using cross-modal pretraining on large-scale video datasets. The distribution of identities and activities in the video dataset may not be representative of the global human population and the diversity in society. The social, gender, racial, and other biases in the dataset could be amplified during pretraining and knowledge distillation. Also, the video dataset may include some private information, which could be vulnerable to dataset extraction attacks [2]. Moreover, our teacher model learns multi-modal grounding via contrastive learning between video and text tokens. However, each text token describes only certain parts of videos. The errors in multi-modal grounding would also be propagated to student models during knowledge distillation, hence we recommend careful use for real-world applications (similar to previous works in video understanding).

## G License

We will publicly release our code and models. We use standard licenses from the community and provide the links below to the license of the datasets, codes, and models we used in the project. For more details, please see the individual link.

**HowTo100M:** Apache

**WikiPedia / Wiki103:** CC BY-SA

**PyTorch:** BSD-style

**Huggingface Transformers:** Apache

**Torchvision:** BSD 3-Clause

**3D-ResNeXt-152:** MIT

**CLIP:** MIT