# OpenReview forum: "VidLanKD: Improving Language Understanding via Video-Distilled Knowledge Transfer"
_NeurIPS.cc/2021/Conference — NeurIPS 2021 Poster_

### Official Review · Reviewer_hPyo · 2021-07-14

**Rating:** 5
**Confidence:** 3

**Summary:**

The paper proposes a new method to improve language understanding with the help of the vision modality. In the proposed method VidLanKD, a teacher model is firstly pre-trained with video-language contrastive loss on Howto100M data. Then the LM part of the teacher model is used to distill knowledge to a student LM model.

**Limitations And Societal Impact:**

The authors adequately addressed the limitations and potential negative societal impact of their work

**Main Review:**

Using vision modality to help language understanding definitely has huge research value. However, I find the paper lacks detailed explanations on some design choices and also on the reasons why it is more preferable compared with Vokenizer.

1. Why do we need a distillation process from the teacher LM to another model? What will happen if we directly pre-train more with only text on the teacher LM pre-trained with image-caption pairs, and fine-tune it on downstream tasks?

2. The paper mentions one limitation of vokenization, images cannot represent words well that require more activity-based commonsense, which is valid. However, how can the proposed method address this problem?

3. Both the teacher model pre-training and the distillation have multiple losses. I feel that the ratio of those losses is a very important hyper-parameter that can significantly affect the performance. For example, for the teacher LM pre-training, since the captions are simply too short and few compared with the unsupervised text, if we don’t set a high coefficient for the contrastive side,  the MLM loss will dominate, which would make the whole learning-from-vision idea pointless. While the paper lacks thorough discussions on this.


**Time Spent Reviewing:**

2

---

> ### Author Response · Authors · 2021-08-10
> **Author Response to Reviewer hPyo**
>
> Thank you for the useful feedback.
>
> **(1) Two-stage pretraining then finetuning**: We conducted experiments on two stage pre-training as suggested.
>
> | Model  |   SST-2 |   QNLI  |  QQP |   MNLI |
> | --- | --- | --- | --- | --- |
> |(a) Text PT             |   88.8 |  84.9   | 85.3  | 77.4|
> |(b) Video PT          |   84.0  |  78.9  |  84.2  | 73.1|
> |(c) Two-stage PT   |  90.3   |  85.0  |  87.2 |  76.9|
> |(d) VidLanKD         |   91.1  |  85.0  |   87.4 |  78.4|
>
> (a) Text PT: model trained only on Wiki103 with MLM.
>
> (b) Video PT: model trained only on HowTo100M with MLM and CT.
>
> (c) Two-stage PT: model firstly trained on HowTo100M (MLM+CT) and then on Wiki103.
>
> (d) VidLanKD: teacher model is trained on HowTo100M (MLM+CT) and then student model is distilled on Wiki103.
>
>
> As shown in the table, while the two-stage pretraining shows better results than text-only or video-only pretraining, VidLanKD still outperforms the two-stage pretraining on GLUE tasks, especially on SST-2 and MNLI. We will add the result in the next version of the paper.
>
> ---
>
> **(2) How can videos help to learn activity-based commonsense better than images?**: Videos can convey more temporal information such as actions/motions. Video captions (e.g., HowTo100M) also have a larger vocabulary coverage than image captions (e.g., CC or SBU) thus more words could be effectively grounded. Therefore, videos can provide richer visual information than images.
> Table 4 (line 288) shows that video-based KD/vokenization outperforms image-based KD/vokenization on GLUE tasks.
>
> In addition, we compare image-based KD and our VidLan KD on PIQA, a physical commonsense reasoning question answering dataset. As shown in the table, while image-based KD helps to improve accuracy from text-only model, our VidLanKD further improves the accuracy.
>
> | Model | PIQA Accuracy |
> | -- | -- |
> | Text only | 56.9 |
> | + Image KD | 58.9 |
> | + VidLan KD (ours) | 60.0 |
>
> We will include the new experiment result in the final version using the extra page.
>
> Moreover, we also provide below two PIQA question examples where our VidLanKD correctly answers and Image KD incorrectly answers. PIQA has a format of choosing an answer among two candidates given context.
>
> Example #1
>
> Context: how to cut wood on a band saw.
> * (a) get the piece of wood you want to cut and put on your safety equipment. start the saw and cut.
> * (b) start the band saw and put your wood on the top. push it through the blade and let it drop to the floor.
>
>
> * Ground-truth answer: (a)
> * Image KD prediction: (b)
> * VidLanKD prediction: (a)
>
> The video clip from HowTo100M (https://www.youtube.com/watch?v=38FqlXKZ6LA) illustrates how to cut wood with a band saw, which helps models to answer the question.
>
> Example #2
>
> Context: how do you properly prepare a steak.
> * (a) take the steak out of warm storage and let come to room temperature, generously add salt and pepper to both sides and let sit for 10 minutes.
> * (b) take the steak out of cold storage and let come to room temperature, generously add salt and pepper to both sides and let sit for 10 minutes.
>
>
> * Ground-truth answer: (b)
> * Image KD prediction (a)
> * VidLanKD prediction: (b)
>
> The video clip from HowTo100M (https://www.youtube.com/watch?v=MMtiszBnpuc) illustrates a brisket recipe where beef is marinated and stored in a 'cold' fridge, which helps our model to answer the question.
>
> We will include such qualitative examples in the final version of the paper.
>
> ---
>
> **(3) Weights for Cross-modal pre-training objectives**:
>
> |                          |  Perplexity   |    Contrastive Loss |    SST-2  | QNLI |   QQP |  MNLI  |
> | -- | -- | -- | -- | -- | -- | -- |
> |CT/MLM = 1:1     |  37.36     |          0.9694         |         91.1  |    85.0 |  87.4 |   78.4 |
> |CT/MLM = 10:1   |  37.97    |           0.9635         |        90.4   |   84.9  | 87.8 |   77.8 |
>
> We use video captions for both CT and MLM objectives for teacher model pretraining. This is illustrated in Sec. 3.2. Moreover, HowTo100M has long videos that involve long captions (>100 tokens on average).
> We perform a new experiment that sets the weight of contrastive learning to 10.0. Increasing the weight of the contrastive learning objective shows almost no effect on the downstream task performance (< 1% on all four GLUE tasks). We will add the experiment results in the next version of the paper.

---

> > ### Comment · Reviewer_hPyo · 2021-08-20
> > **The new experiment results are helpful**
> >
> > I think the new results provided during rebuttal is helpful. I've changed my rating accordingly.
> >
> > I have one extra concern. I noticed that you use triplet loss instead of N-pair loss (https://papers.nips.cc/paper/2016/file/6b180037abbebea991d8b1232f8a8ca9-Paper.pdf) for contrastive learning. While the latter one can be more efficient. Any reasons behind it?  Or do we have some training time complexity analysis?

---

> > > ### Author Response · Authors · 2021-09-01
> > > **Author response to the new reply by Reviewer hPyo**
> > >
> > > Thank you for giving suggestions. We used triplet loss for fair comparison with Vokenization paper [1]. We conducted an ablation experiment using triplet vs. N-pair loss for teacher model  pretraining. For N-pair loss [2], concretely, for every batch with N image-text pairs, we create N positive pairs and use the other N*(N-1) pairs as negative examples.
> > >
> > > |         Model           |   SST-2  |  QNLI  |  QQP  |  MNLI |
> > > | --- | --- | --- | --- | --- |
> > > | KD-NST \w triplet   |91.1    |    85.0  |   87.4  |   78.4 |
> > > | KD-NST \w npair    | 90.6    |   85.7  |  87.6   |  78.3  |
> > >
> > > As shown in the table, so far we have not observed any large differences in finetuning performance on GLUE tasks (<1% accuracy on all 4 tasks) between using triplet and N-pair losses. We will add the result and discussion in the final version of our paper.
> > >
> > > We are also glad that our rebuttal was helpful; hopefully our new answers above will address your remaining questions (we are also happy to answer any other remaining questions) and we hope there can be further updates/increase on your judgment/score of our work. We really appreciate it.
> > >
> > > [1] Hao Tan and Mohit Bansal, "Vokenization: improving language understanding with contextualized, visual-grounded supervision", 2020, in EMNLP.
> > >
> > > [2] Kihyuk Sohn, "Improved deep metric learning with multi-class n-pair loss objective", 2016, in NIPS.

---

### Official Review · Reviewer_X96L · 2021-07-15

**Rating:** 6
**Confidence:** 3

**Summary:**

The paper proposes to improve language understanding using a video-language knowledge distillation method. Compared to the previous state of the art approach, “Vokenization”, they use a large multi-modal video dataset for video-language pre-training to accommodate larger vocabulary needed for better language understanding. For their knowledge distillation based approach, the teacher is the pre-trained model on the multi-modal dataset whose weights are distilled to the student language model trained only on a text dataset. This also avoids the approximation error in the vokenization approach. They perform extensive experiments and improve performance on several downstream language understanding benchmarks (GLUE, SQuAD and SWAG) and also show the commonsense knowledge and reasoning learned from videos.


**Limitations And Societal Impact:**

I recommend the authors to add the broader impact and limitations of the paper from the supplementary to the main paper. The main limitation discussed by the authors is the bias in the video datasets (gender, racial, etc.) and this bias getting amplified during training.
Another potential limitation is the carbon footprint and cost effectiveness of training on these large-scale video datasets. Is there any alternative where you could still use visual information for improving language understanding in a more environmentally friendly way ? An ideal proposition but worth thinking about.


**Main Review:**

The paper is marginally novel and looks at an interesting side of improving language understanding models using video-language based pre-training. It is well written and comprehensive. They perform extensive experiments to show the effectiveness of their method on downstream tasks and several ablation studies to compare different alternatives for the loss functions and pre-training datasets.

1. In line 104, the authors discuss one of the limitations of the vokenization paper - the approximation error and that their method addresses this. It is though not very clear what this error is and how the method takes care of it?

2. Vokenization trains on MSCOCO which is a relatively small image-text dataset, but combining it with Conceptual Captions [1] or SBU captions [2] datasets wouldn’t be enough? What is training on video datasets bringing extra than just on the combination of the above image datasets?

3. What happens if we take the teacher language model after cross-modal pretraining and fine-tune on the text pre-training dataset without knowledge distillation? I believe this is different from the baseline mentioned in Table 1., first row.

4. Why are the ablation experiments done on BERT 6L/512H whereas the main model is trained on BERT 12L/768H ?

5. Although the models show effectiveness over different tasks and datasets, the novelty of the method is limited as it uses state of the art knowledge distillation and pretext self-supervised objectives for training the model.

[1]Sharma, P., N. Ding, S. Goodman, et al. Conceptual captions: A cleaned, hypernym, image-text dataset for automatic image captioning, ACL, pages 2556–2565. 2018
[2] Ordonez, V., G. Kulkarni, T. L. Berg. Im2text: Describing images using 1 million captioned photographs, NIPS 2011



##Post Rebuttal

I have carefully read the authors' response and they address my concerns. I appreciate the new experimental results provided by the authors and they give a better idea of the issue raised. I maintain my original rating.



**Time Spent Reviewing:**

3.5

---

> ### Author Response · Authors · 2021-08-10
> **Author Response to Reviewer X96L**
>
> Thank you for the useful feedback.
>
> **(1) Approximation error**: Using 30K candidate images from MS COCO (mostly natural images) cannot cover diverse real-world text from Wikipedia. The voken supervision depends on the performance of the text-to-image retrieval model. They use a finite number of labels to supervise the text model.
> To alleviate this problem, we adopt knowledge distillation objectives that do not depend on a finite number of labels. For example, CRD [1] learns a representation that is close in some metric space for "positive" pairs and pushes apart the representation between "negative" pairs. NST [2] uses MMD (mean maximum discrepancy) which maps two distributions to a joint space to calculate distance. Since these KD methods distill information from teacher models to student models regardless of the number of labels, they do not suffer from the finite label problem.
>
> ---
>
> **(2) Advantages of video dataset**: Videos can convey more temporal information such as actions/motions. Video captions (e.g., HowTo100M) also have a larger vocabulary coverage than image captions (e.g., CC or SBU) (line 212-215) thus more words could be effectively grounded. Therefore, videos can provide richer visual information than images.
> Table 4 (line 288) shows that video-based KD/vokenization outperforms image-based KD/vokenization on GLUE tasks.
>
> ---
>
> **(3) Two-stage pretraining then finetuning**: We conducted experiments on two-stage pre-training as suggested.
>
> | Model  |   SST-2 |   QNLI  |  QQP |   MNLI |
> | --- | --- | --- | --- | --- |
> |(a) Text PT             |   88.8 |  84.9   | 85.3  | 77.4|
> |(b) Video PT          |   84.0  |  78.9  |  84.2  | 73.1|
> |(c) Two-stage PT   |  90.3   |  85.0  |  87.2 |  76.9|
> |(d) VidLanKD         |   91.1  |  85.0  |   87.4 |  78.4|
>
> (a) Text PT: model trained only on Wiki103 with MLM.
>
> (b) Video PT: model trained only on HowTo100M with MLM and CT.
>
> (c) Two-stage PT: model firstly trained on HowTo100M (MLM+CT) and then on Wiki103.
>
> (d) VidLanKD: teacher model is trained on HowTo100M (MLM+CT) and then student model is distilled on Wiki103.
>
> As shown in the table, while the two-stage pretraining shows better results than text-only or video-only pretraining, VidLanKD still outperforms the two-stage pretraining on GLUE tasks, especially on SST-2 and MNLI. We will add the result in the extra page of the final version of the paper.
>
> ---
>
> **(4) Ablation experiments on small models**: Due to limited computational resources in academia, we had to run ablation experiments on the smaller model. Other papers also perform ablation on different scales of model [3].
>
> ---
>
> **(5) Novelties**: We introduce a video-based knowledge distillation method for LMs to improve NLU. We provide the viewpoint of vokenization as a knowledge distillation and experiment with different objectives to overcome its limitation of using finite labels. We show video-based KD could be more effective than image-based KD on many NLU benchmarks. We also provide detailed ablations of the reason for the improvements and analysis on the knowledge learned from videos including linguistic knowledge and physical/temporal commonsense.
>
> ---
>
> **(6) Broader impact and limitations**: We placed the broader impact and limitations section in the supplementary due to page limitations. We will move the section to the main paper using the extra page allowed for the camera-ready version.
>
> ---
>
> **(7) More environmentally friendly setup**: In our experiments, we use the same architecture for teacher and student models. To reduce carbon footprint, we can distill the cross-modal knowledge into a smaller model that processes text more efficiently with less energy consumption.
>
> ---
>
> References (also all included in original submission):
>
> [1] Yonglong Tian, Dilip Krishnan, and Phillip Isola. 2020. Contrastive representation distillation. In ICLR.
>
> [2] Zehao Huang and Naiyan Wang. 2017. Like what you like: Knowledge distill via neuron selectivity transfer. arXiv preprint arXiv:1707.01219.
>
> [3] Hao Tan and Mohit Bansal. 2020. Vokenization: improving language understanding with contextualized visual-grounded supervision. In EMNLP.

---

### Official Review · Reviewer_ZhJy · 2021-07-16

**Rating:** 7
**Confidence:** 3

**Summary:**

The paper studies the problem of how to use visual information to help NLP models. It argues that videos contain rich visual grounding information and could potentially benefit pure-text NLP tasks. It proposes to distill such visual information into a language model during pre-training. Experiments on GLUE benchmark suggest the efficacy of the method.

Overall, I applaud the novelty and problem the paper tries the address. However, the experiments could be strengthened by better control of the baselines and more analysis (see cons). I lean towards acceptance but I would suggest adding the suggested experiments later if possible.

**Limitations And Societal Impact:**

Nothing I could think of.

**Main Review:**

Pro:
The paper tackles a long-standing problem: grounding NLP models. The paper takes a practical yet interesting angle: distill grounding information in the pre-trained language models so it could benefit various NLP tasks.

The distill approaches make sense and works well in experiments.



Con:

1. Potentially important baselines missing. To verify that the gain is from injected visual information, the authors introduces a student model that is pre-trained on HowTo100M. However, another possible contributing factor is the distillation process, as the distillation process itself might bring performance improvement, as long as the teacher is a reasonable model.
Thus, I think a baseline where the teacher is trained only on the text of HowTo100M/Wiki and distilled to a student should be included to disentangle the effect of the visual information and the effect of the distillation process.

2. Concern over the setting. Is the teacher model initialized from BERT? If so, then a baseline where the teacher is simply the original BERT should be included, to investigate if some of the useful information distilled into the student is from BERT.

3. Lack of analysis. While showing improvement on GLUE benchmarks is a good sign, it is still hard to see how the grounding information helps a concrete NLP task (e.g., it might be hard to imagine how exactly videos help SST-2). Some examples of how the grounding helps PIQA or Knowledge test in GLUE could be helpful.
Another potential aspect I could think of is to test whether the grounded pre-training relieves reporting bias [1,2].
I won’t count this as a reason for rejection but would appreciate a better analysis.
[1] Reporting bias and knowledge acquisition. AKBC 2013.
[2] Do Neural Language Models Overcome Reporting Bias? COLING 2020.


**Time Spent Reviewing:**

3

---

> ### Author Response · Authors · 2021-08-10
> **Author Response to Reviewer ZhJy**
>
> Thank you for the useful feedback.
>
> **(1) ​​Potentially important baselines missing**: We made this comparison in supplementary Table 7, "+KD from $T^{MLM}$", which refers to the student model performance distilled from the teacher model trained only on MLM (masked language modeling). This means, in this setting, the teacher model is only trained on HowTo100M captions, without using videos. As shown below, distillation from the teacher trained with both video and captions ($T^{MLM+CT}$) outperforms the distillation from teacher only trained with captions ($T^{MLM}$) on 4 GLUE tasks.
>
> |                        | SST-2 |  QNLI |  QQP  |  MNLI |
> | -- | -- | -- | -- | -- |
> | $T^{MLM}$         |  88.1   |   83.1  |   85.6  |  77.4 |
> | $T^{MLM+CT}$  |  91.1  |    85.0   |  87.4 |   78.4 |
>
> We will move this to the main paper using the extra page provided for camera-ready.
>
> ---
>
> **(2) Weight initialization**: In lines 135 and 155, we discuss that both the teacher and student model are trained from scratch.
>
> ---
>
> **(3) Examples of how visual grounding can help PIQA**: Below we show two questions from PIQA and two video clips from HowTo100M, where VidLanKD can help the language model to answer questions. PIQA has a format of choosing an answer among two candidates given context, which requires physical commonsense reasoning.
>
> Example #1
>
> Context: to remove a screw from a board,
> * (a) place the tip of the screwdriver into the top of the screw and twist in a clockwise direction.
> * (b) place the tip of the screwdriver into the top of the screw and twist in a counter clockwise direction.
>
>
> * Ground-truth answer: (b)
> * Text-model prediction: (a)
> * VidLanKD prediction: (b)
>
> The video clip from HowTo100M (www.youtube.com/watch?v=ASjB-GtyIZE) illustrates how to fix a car cup holder that involves removing a screw with a screwdriver, which helps models to learn the action of how to remove a screw from another object.
>
> Example #2
>
> Context: how to grow a plant.
> * (a) bury seed in sand and add 1 cup of water daily.
> * (b) bury seed in soil and add 1 cup of water daily.
>
>
> * Ground-truth answer: (b)
> * Text-model prediction (a)
> * VidLanKD prediction: (b)
>
> From this video clip from HowTo100M (www.youtube.com/watch?v=NQCuOKFwQ4Q), the model can learn from the visual of planting in soil, which helps models to identify the correct action on planting.
>
> We will include more such qualitative examples in the final version of the paper.
>
> ---
>
> **(4) Reporting bias**: We conducted the color prediction experiment following the COLING 2020 paper, thanks for the useful suggestion. While LMs are to some extent capable of learning association between concepts and their properties indirectly by aggregating across contexts, during this process, they often overgeneralize, predicting semantically similar but mutually exclusive values. To test this over-generalization phenomenon, the authors evaluate LMs' ability to predict colors from objects. The authors constructed a list of 11 common colors and extracted all sentences in Wikipedia in which a color modifies a noun, masking the color tokens (e.g. "A bear is [MASK]").
>
> In the table below, we show the accuracy of the text-only baseline and our KD approach on the color prediction task on Zero-shot and Fine-tuned settings. We can see that video grounding can help models better associate concepts with colors (alleviating reporting bias) in both settings.
>
> |Model                |      Zero-Shot  |    Fine-tuned |
> | -- | -- | -- |
> |BERT$_{6L/512H}$   |          11.81    |             62.08 |
> |+ KD-NST           |         12.29   |             63.40  |
>
> We will include this result using the extra page in the final version.

---

> > ### Comment · Reviewer_ZhJy · 2021-08-26
> > **Response**
> >
> > Thank you for pointing out the results in the appendix. Now all of my concerns are resolved.
> >
> > One suggestion is that I think the result of “distilling from a vision-and-language model gives an evident improvement over distilling from a pure-language model” (Sup Table 7) is a crucial point to support the general claim of “visual supervision enhances NLP models”. Thus, it might be better to highlight this point in the paper.

---

> > > ### Author Response · Authors · 2021-09-01
> > > **Author response to the new reply by Reviewer ZhJy**
> > >
> > > Thank you for the suggestion.
> > >
> > > We will move the ablation results in the main paper and highlight the point in the final version.

---

### Official Review · Reviewer_aMi4 · 2021-07-16

**Rating:** 7
**Confidence:** 4

**Summary:**

The paper proposed to distill the large-scale pre-trained video-language multi-modal knowledge into the language modeling. Different knowledge distillation objectives are applied. In the downstream task, the distilled language model shows performance improvement over the conventional language model.

**Ethics Review Area:**

["I don’t know"]

**Limitations And Societal Impact:**

There is no potential negative societal impact of their work.

**Main Review:**

Strength:
+ The novelty comes from dIstilling the video-language multimodal knowledge learned by video-text pre-training into the language modeling. That could enable the language model the ability to deal with more visual-relevant commonsense such as action.
+ Multiple knowledge distillation methods are well studied and compared.
+ Proposed method outperforms the conventional language model pre-trained only by text in quite a few downstream tasks.
+ The fair comparison against the Vokenization method further validates the effectiveness of the proposed framework.

Weakness:
- When comparing with Vokenization, 30K videos are pre-selected according to L205. A question about the detail is how to select the videos?
- In the downstream tasks, it is interesting to see in what samples your model is generally better than text-pretrained language model.  Does that correspond to your motivation that your model should help in activity-based and physical commonsense knowledge?

Clarity:
The paper is clearly illustrated and well written.

**Time Spent Reviewing:**

3

---

> ### Author Response · Authors · 2021-08-10
> **Author Response to Reviewer aMi4**
>
> Thank you for the useful feedback.
>
> **(1) More details about video voken sampling**: We first select a video for each of 23K visual tasks. For the remaining 7K vokens, we randomly select 7K visual tasks, then select a video from each visual task. We select videos with lengths ranging from 1 to 20 seconds. More details can be found in supplementary "A.1  Video Voken Sampling".
>
> ---
>
> **(2) Examples where VidLanKD is better than a text-based model on activity-based / physical commonsense dataset**: We show two questions from PIQA, a physical commonsense reasoning dataset, and related video clips from HowTo100M that could help models to answer the questions. PIQA has a format of answering questions by choosing an answer among two candidates given context.
>
> Example #1
>
> Context: to remove a screw from a board,
> * (a) place the tip of the screwdriver into the top of the screw and twist in a clockwise direction.
> * (b) place the tip of the screwdriver into the top of the screw and twist in a counter clockwise direction.
>
>
> * Ground-truth answer: (b)
> * Text-model prediction: (a)
> * VidLanKD prediction: (b)
>
> The video clip from HowTo100M (www.youtube.com/watch?v=ASjB-GtyIZE) illustrates how to fix a car cup holder that involves removing a screw with a screwdriver, which helps models to learn the action of how to remove a screw from another object.
>
> Example #2
>
> Context: how to grow a plant.
> * (a) bury seed in sand and add 1 cup of water daily.
> * (b) bury seed in soil and add 1 cup of water daily.
>
>
> * Ground-truth answer: (b)
> * Text-model prediction (a)
> * VidLanKD prediction: (b)
>
> From this video clip from HowTo100M (www.youtube.com/watch?v=NQCuOKFwQ4Q), the model can learn from the visual of planting in soil, which helps models to identify the correct action on planting.
>
> We will include more such qualitative examples in the final version of the paper.

---

### Decision · Program_Chairs · 2021-09-27

**Decision:**

Accept (Poster)

**Comment:**

All but one reviewer recommend the acceptance of this paper.

The reviewer not recommending acceptance has remaining concerns about positioning this work with respect to vtokenizer and the AC encourages the authors to address this. Please also be careful about explaining additional details in this submission as requested by this reviewer. However other reviewers were quite positive about this work and one reviewers noted that they felt that this paper does indeed make a more convincing case for the general topic of creating visually-grounded NLP models than Vokenization.

The AC recommends acceptance.